# A 5-Year Study of Antiseizure Medications (ASMs) Monitoring in Patients with Neuropsychiatric Disorders in an Italian Clinical Center

**DOI:** 10.3390/ph16070945

**Published:** 2023-06-29

**Authors:** Letizia Biso, Marco Carli, Shivakumar Kolachalam, Giorgio Monticelli, Pasquale Fabio Calabrò, Antonello di Paolo, Filippo Sean Giorgi, Guido Bocci, Marco Scarselli

**Affiliations:** 1Department of Translational Research and New Technologies in Medicine and Surgery, University of Pisa, 56126 Pisa, Italy; l.biso@studenti.unipi.it (L.B.); carlimarco@outlook.it (M.C.); g.monticelli1@studenti.unipi.it (G.M.); calabro.pf@gmail.com (P.F.C.); filippo.giorgi@unipi.it (F.S.G.); 2Aseptic Pharmacy, Charing Cross Hospital, Imperial College Healthcare NHS Trust, London W6 8RF, UK; k.shivakumar@alumni.sssup.it; 3Department of Clinical and Experimental Medicine, University of Pisa, 56126 Pisa, Italy; antonello.dipaolo@unipi.it (A.d.P.); guido.bocci@unipi.it (G.B.)

**Keywords:** antiseizure medications, valproic acid, levetiracetam, therapeutic drug monitoring, subtherapeutic

## Abstract

Despite receiving appropriate antiseizure medications (ASMs), a relevant percentage of neuropsychiatric patients do not benefit from this approach, and one reason is subtherapeutic ASMs plasma concentration (C(p)) due to improper drug adherence, interindividual pharmacokinetic differences, or metabolic interactions among different drugs. For these reasons, therapeutic drug monitoring (TDM) by measuring ASMs C(p) is an effective tool that improves pharmacological therapies in clinical practice. Based on these premises, in the present real-world study, we analyzed the C(p) of the most used ASMs in diverse medical conditions, which were assayed during the years 2018–2022 at the University Hospital of Pisa, including about 24,000 samples. This population was largely heterogeneous, and our database did not contain clinical information about the patients. The most used ASMs were Valproate (VPA: 54.5%) and Levetiracetam (LEV: 18.6%), followed by Oxcarbazepine (OxCBZ: 8.3%) and Carbamazepine (CBZ: 7.2%), whereas the associations LEV/VPA, Ethosuximide (ESM)/VPA, and CBZ/VPA were the most frequently proposed. In about 2/3 of assays, ASMs C(p) was in range, except for VPA, which was underdosed in almost half of the samples. Importantly, toxic levels of ASMs C(p) were found very rarely. For VPA, there was a decrease of mean C(p) across ages, from adolescents to older patients, while the C(p) of LEV, CBZ, OxCBZ, and Topiramate (TPM) showed a slight tendency to increase. When we compared females and males, we found that for VPA, the average age was higher for females, whereas women taking Lamotrigine (LTG) and OxCBZ were younger than men. Then, comparing ASMs used in neurologic and psychiatric disorders, based on the request form, it emerged that the mean C(p) of CBZ, OxCBZ, and LTG on samples collected in the Psychiatric Unit was lower compared to the Neurology and Child Neuropsychiatry Units. Finally, the ASMs subjected to multiple dosing starting from an initial subtherapeutic C(p) increased their level at different time points within a year, reaching the reference range for some of them. In conclusion, the present study suggests that TDM is widely applied to monitor ASMs C(p), finding many of them within the reference range, as a demonstration of its utility in clinical practice.

## 1. Introduction

Despite receiving the most appropriate pharmacological treatments, more than 1/3 of patients affected by neuropsychiatric disorders do not sufficiently benefit from this approach. This lack of therapeutic efficacy might be due to several reasons, among which are particularly worthy of consideration are an improper drug adherence, the pharmacokinetic interactions between drugs given in combination, and the interindividual differences affecting absorption and metabolism [1,2,3].

For addressing these issues, therapeutic drug monitoring (TDM) is generally considered as an effective and relatively easy tool to improve current pharmacological therapies, as it provides a reference range for the plasma drug concentration (C(p)) with the highest probability of optimal therapeutic response combined with a reduced risk of adverse drug reactions and toxicity [4]. A proper use of TDM should provide important advantages in terms of cost benefits, especially in reduced length of hospital stay and healthcare costs, although controlled studies relative to the cost-effectiveness of TDM are still few, underscoring the need for more evidence on this matter [5].

Unfortunately, TDM is often misused, and its improper application, such as measuring C(p) not at steady-state conditions or at the wrong time during the day, might lead to misleading results and wrong clinical decisions [6].

Out of the several medications used to treat Central Nervous System (CNS) diseases, antiseizure medications (ASMs) are probably the largest category of drugs for which TDM is performed [7].

ASMs are generally used in all the different forms of epilepsy, but some of them are also frequently used for other disorders, primarily in bipolar disorders and secondarily in migraine and neuropathic pain [8,9,10,11]. Dosages of ASMs can be different according to the medical condition [4,12,13]. Importantly, the reference ranges of ASMs C(p) were determined in epileptic patients, whereas TDM studies for the other medical conditions, such as bipolar disorders, are less numerous [13,14,15]. However, according to the recent consensus guidelines for TDM in neuropsychopharmacology, the same reference ranges of ASMs for epilepsy and bipolar disorder have been adopted [4]. Studies on TDM for ASMs used in migraine or neuropathic pain are still missing, limiting their use in clinical practice, so we can assume that in our sample, it was less represented [12,16].

According to the current clinical practice, for focal onset seizures, Carbamazepine (CBZ), Lamotrigine (LTG), and Levetiracetam (LEV) are considered among the first-line treatments, whereas for generalized seizures, Valproate (VPA) is proposed as one of the best options, followed by LTG and LEV [17,18].

However, in the last decade, especially after warnings by regulatory agencies, there is a strong suggestion not to choose VPA in females with childbearing potential [19]. Oxcarbazepine (OxCBZ) is considered a good replacement for CBZ for its better pharmacokinetic profile [20], while for the absences, besides VPA, Ethosuximide (ESM) is considered another valuable option.

Interestingly, the traditional ASMs Phenobarbital (PB) and Phenytoin (PHT), even if they are not considered as a first choice in the US and Europe anymore due to the well-known severity of their side effects, are still largely utilized, mainly in low- and middle-income countries [21,22,23]. Among the ASMs of the new generation, the use of LEV is growing continuously as monotherapy or in association with other ASMs due to its well-demonstrated efficacy in post-marketing studies, tolerability, and lower metabolic interactions with other drugs. Indeed, polytherapy, i.e., the use of two or even three ASMs in the same individual, is not rarely used in clinical practice for patients with drug-refractory epilepsy (DRE), which is a form of epilepsy with uncontrolled seizures that persist after treatment with at least two appropriately chosen and administered ASMs [1,18,24]. The newer ASMs have a better pharmacokinetic profile that could limit drug–drug interference; however, post-marketing data on their efficacy are much fewer compared with LEV, even though they seem promising [25].

In addition, some ASMs, such as VPA, CBZ, OxCBZ, and LTG, are largely used in bipolar disorders, especially VPA, which is considered, together with lithium, a cornerstone for the treatment of these psychiatric conditions [26].

Several reasons can explain the frequent use of TDM for ASMs, compared with other classes of CNS drugs. These include the difficulty to predict the onset of seizures; the need to confirm patients’ adherence to the treatments in the case of seizure recurrence; and the need to reveal the subtherapeutic C(p) that can be responsible for ASM inefficacy in patients fully compliant, or allows recognizing if signs of toxicity are related to high C(p) [27,28]. On this matter, subtherapeutic levels are often found in patients accessing the hospital for seizure recurrence [29].

In addition, considering that epilepsy is particularly common in newborn/children and older people, these special categories of patients may particularly benefit from TDM due to their particular pharmacokinetic characteristics in terms of drug metabolism and elimination [30]. In a way, the same concept applies to pregnant women, where the serious risk of teratogenicity induced by ASMs, particularly the older ones, and especially VPA, imposes a frequent monitoring of drug levels, when these drugs cannot be avoided [31]. Clinical studies analyzing correlations between C(p) and ASMs clinical efficacy have found good evidence for PHT, CBZ, and in part VPA, while for the other ASMs, data are still inconclusive [4,32].

Moreover, ASMs are subject to relevant pharmacokinetic interactions when given in association together or with other drugs strongly influencing their C(p), even if used at standard dosage. On this issue, we should remind readers that some classical ASMs are powerful CYP enzymes inducers or inhibitors, where CBZ, PHT, and PB mostly induce C(P)Y3A4, CYP1A2, and CYP2C9, while VPA inhibits CYP2C9, CYP2C19, and UGT enzymes [33,34].

Based on these premises, in this study, we analyzed the C(p) of the most used ASMs in diverse medical conditions during the years 2018–2022 at the University Hospital of Pisa, including about 24,000 samples. We did not have clinical information about the patients, so the sample was largely heterogeneous. This analysis has taken into considerations the relevance of gender, age, and other parameters to examine their influence on ASMs C(p) values. Importantly, patients monitored for repeated dosing at different time points within a year were included in our evaluation. Finally, we tried to compare the C(p) used in neurologic and psychiatric patients, based on the request form coming from the Neurology or Psychiatry Unit that allowed us to make this comparison.

## 2. Results

### 2.1. Description of ASMs C(p) Samples during the Period 2018–2022

The total number of samples was 23,946, distributed as follows: 13,039 samples of VPA, 4453 of LEV, 1988 of OxCBZ, 1730 of CBZ, 727 of LTG, 576 of ESM, 560 of PB, 499 of PHT, 257 of Topiramate (TPM), and 116 of Lacosamide (LCS) (Figure 1A). From this analysis, we decided to exclude ASMs with less than 100 samples (i.e., zonisamide).

The internal requests for blood samples were coming from the Unit of Psychiatry (6064), Child Neuropsychiatry (4092), and Neurology (1036), whereas the other requests (12,753) were from other units of the University Hospital of Pisa, or they were external requests coming from the northwest area of the Tuscany region (Figure 1B).

The number of requests varied during these years, with 5364 samples analyzed in 2018, 4846 in 2019, 4099 in 2020, 5133 in 2021, and 4503 in 2022 (until November). We hypothesize that the number of requests dropped slightly during the year 2020 due to restrictions in healthcare services for the COVID-19 pandemic that began to spread in Italy during that year. The total number of patients was 7764, of which 3765 (48.5%) were females and 3999 (51.5%) were males, with a mean age of 49.2 ± 24.8 years. Interestingly, for ESM, the mean age was 15.7 years, while for LEV and PHT, they were 61.7 and 63.7 years, respectively, suggesting that in our sample, we found the use of some ASMs in specific categories of the general population.

**Figure 1 pharmaceuticals-16-00945-f001:**
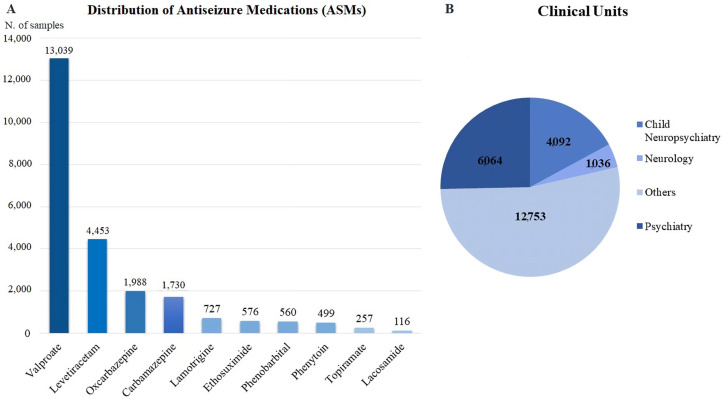
(**A**) Distribution of the antiepileptic drugs (ASMs) among the total blood samples recorded at the University Hospital of Pisa from January 2018 to November 2022. (**B**) Distribution of ASMs blood samples in the different clinical units (Child Neuropsychiatry, Neurology, Psychiatry, and other units).

### 2.2. General Distribution of ASMs C(p) Samples

Overall, about 2/3 of the reported ASMs C(p) values were within the reference range, with the exception of VPA (49.8%), TPM (55.2%), and PHT (56.7%), where this was the case for only about half of the samples (Table 1).

Considering VPA, it was below the range in 48.4% of the C(p) samples, while it was above the range only in 1.8%, and at toxic concentrations in 0.2% (>120 μg/mL). On the contrary, 32.2% of the C(p) samples of TPM was found above the recommended range, with 5.8% above the toxicity level of 16 μg/mL, while the C(p) of PHT was largely underdosed (30.2%). It is important to notice that 56.7% of patients who were taking PHT were over 65 years of age, and this could explain such a trend. The two ASMs with the highest number of samples within the therapeutic range were PB (85.6%) and LCS (85.1%).

In general, most of the ASMs showed a quasi-gaussian distribution, with a peak within the therapeutic range; however, since they failed the Kolmogorov–Smirnov normality test, non-parametric statistics tests were used in most cases (Figure 2).

### 2.3. Distribution of ASMs C(p) Samples across Different Ages

In relation to C(p) samples’ distribution across different ages, we performed one-way ANOVAs or the Kruskal–Wallis test according to data distribution to check for each ASM for significant differences among 3 age groups, arbitrarily divided into “under 18 years”, “between 18–65 years”, and “over 65 years” (Figure 3). In the table, it is also reported the number of ASM samples in parentheses for each subgroup. For ESM and LCS, the Mann–Whitney test was performed due to the limited number of samples in certain age groups. We also performed multiple comparison analysis (either Tukey’s or Dunn’s, depending on ANOVA results) to assess differences between each age group (Figure 3).

For VPA, we found a statistically significant difference among the 3 groups’ C(p) (*p* < 0.001, F = 12.88), with the group under 18 years being the one with the highest mean C(p) (61.4 ± 20.7 μg/mL), and the group over 65 years being the one with the lowest mean C(p) (42.4 ± 20.7 μg/mL). Post hoc analysis confirmed a significant difference between each group (*p* <.001).

There was also a significant difference across different ages for LEV (*p* < 0.001, H = 63.73), but with an opposite trend compared to VPA, with the eldest group being the one with the highest mean C(p) (24.9 ± 17.6 μg/mL). Post hoc analysis showed a statistically significant difference between the eldest group and the 18–65 years group (*p* < 0.001).

Similar to LEV, OxCBZ and CBZ C(p) were the lowest in the under 18 years group (12.6 ± 5.1 μg/mL and 6.6 ± 2.17 μg/mL, respectively). For OxCBZ, there was a statistically significant difference among the under 18 years group (*p* = 0.006) and the 18–65 years group or the over 65 years group (*p* = 0.003), while no difference was found between the last two groups. Similarly, for CBZ, multiple comparison analysis showed a difference only between the under 18 years and the 18–65 years group (*p* = 0.03) or the over 65 years group (*p* = 0.007). With PB, there was just a statistically significant difference between the under 18 years group and the 18–65 years (*p* = 0.02), while no difference was found among the other groups.

We also found for TPM and LCS, a difference between the 18–65 years and the over 65 years group (*p* < 0.001), with the second group with the highest mean C(p), similarly to other ASMs considered in this study. Finally, we did not find any differences among the three groups for LTG, ESM, and PHT.

To further analyze the relationship between age and dosage, we performed Spearman r correlations between the two variables for each ASM (Figure 4). For VPA, the Spearman r test confirmed a negative correlation (r = −0.29; *p* < 0.001) between age and VPA dosage, while a positive correlation (r =0.27; *p* < 0.001) was observed for LEV and for OxCBZ (r = 0.08; *p* = 0.002). PB samples showed a negative correlation between age and dosage (r = −0.14; *p* = 0.0014), while a positive correlation was found for LCS (r = 0.41; *p* < 0.001). The other ASMs did not show statistically significant correlations.

### 2.4. Distribution of ASMs C(p) Samples in Relation to Gender

When analyzing age differences, we found that females taking VPA or LCS were significantly older than males (48.9 ± 23.6 vs. 40.2 ± 22.5 years, respectively), while males taking OxCBZ, LTG, or PB were older than females (Table 2). The other ASMs did not show relevant differences when comparing mean ages in the two genders.

Then, we analyzed if there was any difference related to gender in terms of ASMs mean C(p), and we found only small differences for LEV, OxCBZ, and CBZ by performing a *t*-test among the groups.

For LEV, the C(p) of females was slightly higher than in males, while it was the opposite for OxCBZ and CBZ. For all the other ASMs, we did not find any statistically significant differences, indicating that gender was irrelevant in the C(p) distribution.

Interestingly, and as expected, the number of males taking VPA was higher than females (6888 vs. 5635), while the number of females taking LTG was 2.5 times more than males (502 vs. 206).

### 2.5. Analysis of the ASMs C(p) Samples Comparing the Neurological, Psychiatric, and Child Neuropsychiatry Units

For each of the four ASMs (VPA, CBZ, OxCBZ, and LTG) that are used for treating both neurological and psychiatric disorders, we calculated the C(p) in samples collected in each of the three Pisa University Hospital Units in order to see whether there was any difference among these specific groups. The numbers of samples are reported in Table 3. For CBZ, OxCBZ, and LTG, we found a higher mean C(p) for the samples coming from the Neurology Unit compared to the Psychiatric Unit. In the case of the Child Neuropsychiatry Unit, the mean C(p) of OxCBZ and LTG was similar to the Neurology Unit. It is worth noting that the samples coming from the Child Neuropsychiatry Unit refer to either neurological or psychiatric patients.

When we analyzed VPA samples, we found a slight difference between the C(p) coming from the Psychiatry Unit compared to the Neurology Unit, where the mean C(p) measurements were 51 ± 21.2 ug/mL and 45 ± 22.4 ug/mL, respectively. However, it should be noted that the patients of the Neurology Unit had a mean age that was higher than the psychiatric one, 63.2 years and 49.1 years, respectively. On the contrary, the mean C(p) of the Neuropsychiatric Unit, as mentioned before, was higher than the other two units (60.9 ± 20.6 ug/mL).

### 2.6. Analysis of Multiple Dosing for Patients Plasma Monitored Repeatedly during One Year

Since ASMs are medications generally used in chronic disorders, patients often monitor their C(p) several times during the treatment. Thus, in another analysis, for each ASM, we selected patients that were monitored at least 3 times (T0–T2) in a 12-month period, with an interval among the samples of at least 1 week, and we compared the C(p) found at different time points. We also performed the Friedman test and multiple comparison analysis when *p* values for the Friedman test were significant.

For most ASMs, we did not find any significant differences among the three C(p) measurements within the year, demonstrating the stability of measurements during chronic treatment (Figure 5A). Interestingly, when the initial mean value (T0) of the ASM C(p) was above the range, i.e., for TPM, or below it, i.e., for LCS, there was a clear change in the following C(p) measures to correct the first concentration, although they were not statistically significant for the limited number of the samples. The ASMs included in this analysis were VPA (*n* = 1066), LEV (*n* = 276), OxCBZ (*n* = 38), CBZ (*n* = 112), LTG (*n* = 49), ESM (*n* = 62), PB (*n* = 41), PHT (*n* = 26), TPM (*n* = 15), and LCS (*n* = 4).

Then, we considered the specific subgroup of patients, who were monitored three times within the year, where the initial ASM C(p) (T0) was below the recommended range (this sub-analysis was conducted only for ASMs with at least 10 samples below range). The ASMs included in this analysis were VPA (*n* = 500), LEV (*n* = 84), LTG (*n* = 18), and ESM (*n* = 11).

We performed Friedman tests and post hoc analysis when needed, and we found that concerning all four ASMs taken into consideration, there was a significant change between T0 and T1/T2 values, which suggests that retesting of blood concentrations was likely conducted after dosages adjustment (Figure 5B).

**Figure 5 pharmaceuticals-16-00945-f005:**
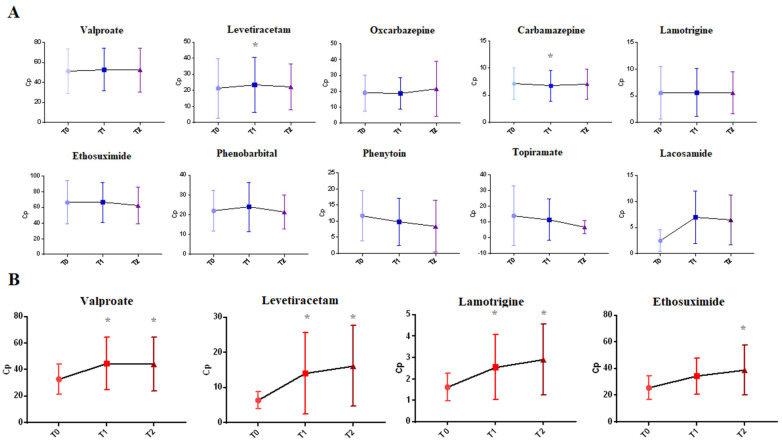
(**A**) ASMs C(p) in patients receiving three consecutive dosings within one year (T0, T1, and T2). * Statistically significant differences in the groups compared to T0 (*p* = 0.003 for LEV; *p* = 0.03 for CBZ). (**B**) ASMs C(p) in patients with subtherapeutic C(p) (red) at T0 (four ASMs) receiving three consecutive dosings within one year (T0, T1, and T2). * Statistically significant differences in the groups compared to T0 (*p* < 0.001 for VPA and LEV; *p* = 0.001 for LTG; *p* = 0.01 for ESM).

### 2.7. Analysis of the ASMs in Combination Therapy

From the analysis of the data reported on the request forms (i.e., taking requests where different ASMs were monitored simultaneously), approximately 10% (686 out of 7764) of patients included in the analysis were taking ASMs in association. However, this number could represent an underestimation considering that some associations could have been missed because the measurement of each ASM taken in combination was made at different time points and not revealed by the request form.

Of these 686 patients taking combination therapy, 586 patients (85.4%) were taking 2 ASMs, 86 patients (12.5%) were taking 3, and 14 patients (2.1%) were taking more than 3. The most common combinations were LEV/VPA (106), ESM/VPA (73), and CBZ/VPA (65). Other combinations were found reflecting personalized therapies according to patients’ characteristics (Figure 6).

## 3. Discussion

In our study, we analyzed the results of plasmatic levels of the ten most used ASMs in diverse neuropsychiatric conditions that were performed at the University Hospital of Pisa as internal and external requests. More than half of the samples included VPA (54.5%), which resulted by far as the most frequently tested ASM. This confirms that TDM for VPA is a standardized practice in clinical settings. This drug has been used for several decades for treating a variety of neurological and psychiatric conditions. VPA represents, for many neurologists, the gold standard in terms of efficacy for treatment of several forms of epilepsy, such as epilepsy with generalized tonic–clonic seizures, and juvenile myoclonic and absence epilepsy; it is considered effective also for focal seizures, and in addition, it has also found an established role in the treatment of BD [35,36,37,38,39]. However, its serious adverse effects impose a constant monitoring of patient’s conditions, including C(p) measuring [7]. VPA utilization has become a standard approach for treating particular categories of bipolar disorders, such as mixed states, rapid cycling, and with anxiety in comorbidity [40]. In addition, VPA can be also used in migraine; however, data of TDM for this medical condition are still missing, so we can assume that in our sample, it was less represented [41].

Interestingly, the second ASM assayed was LEV (18.6%), a second-generation drug used to treat different forms of epilepsy [42,43]. A recent network meta-analysis has suggested the use of LEV as a first-line treatment for focal and generalized seizures, especially in patients where VPA is not recommended [17]. The success of this relatively new ASM is due to its efficacy, tolerability, and very limited pharmacokinetic interactions with other medications. Although LEV is approved for treating epilepsy at all ages, in our sample, the mean age of patients taking this drug was quite high (61.9 years), and nearly half of these patients were over 65 years. This might be explained by the caution regarding its use, especially at high dosages, in younger patients due to the risk of behavioral and psychiatric adverse reactions, discovered a few years after its post-marketing introduction, and now widely acknowledged among neurologists [44,45]. On the contrary, for patients under 18 years, besides VPA, ESM was often used, which is known to be efficacious in the absences. In addition, the neuropsychological issues in child epilepsy should be taken into serious consideration [46].

CBZ and OxCBZ together were assayed in the remarkable amount of almost 4000 samples (15.5%). This may indirectly confirm not only the still frequent use of CBZ, a well-known efficacious ASM used in several forms of epilepsy, particularly for focal seizures, but also how TDM of this drug is considered useful by many physicians. OxCBZ, a cognate compound of CBZ, shares a similar mechanism of action, but with a lower degree of metabolic interactions with ASMs and other drugs.

The next most frequently tested ASM in our case series was LTG with 727 samples, followed by ESM, PB, PHT, TPM, and LCS. Some studies have indicated LTG as a first-choice option for focal and generalized seizures, and this ASM is very popular, especially in the UK [17,18,47,48].

Probably, its lower number of samples compared to the most used ASMs is due to the need of a slow titration to reach the therapeutic C(p), along with some concerns for its rare, but dangerous, skin-related side effects, such as the Stevens–Johnson syndrome, which makes its use less diffuse in Italy [49,50].

When we analyzed the ASMs C(p), about 2/3 of these blood samples were in range, demonstrating that many clinicians are making the effort to monitor C(p) in epileptic patients. Differently from most ASMs, for VPA, almost half of the samples revealed underdosage, with a mean C(p) of 51.3 μg/mL. However, when we compared the mean VPA C(p) distribution across different age groups, we found significant differences, with the lowest mean C(p) in the elderly (42.4 μg/mL) and the highest in adolescents (61.4 μg/mL). This might reflect a tendency of being more careful with the dosage in older people and/or a more precise drug administration in children and adolescents involving parenting control [51,52,53].

Similar to VPA, the percentage of C(p) samples below the therapeutic range for PHT and LTG were 30.2% and 32.8%, respectively. For PHT C(p) assays, the higher mean age of patients (63.7 years, with 56.7% of patients being over 65 years) could possibly be in line with a more cautious way of administering this drug to the elderly. In contrast, in about 1/3 of TPM assays, its levels were above the recommended C(p). TPM should be closely monitored, especially in the elderly, as it has shown negative effects on cognition and memory [54].

If, for VPA, there was a decrease of the mean C(p) from adolescents to older patients, for the other ASMs, we found two different trends: there was either no change, or there was an increase of C(p) with age. More in detail, the C(p) of LTG, PB, PHT, and ESM was stable across ages, while the C(p) of LEV, CBZ, OxCBZ, TPM, and LCS showed a tendency to increase.

The higher mean C(p) of LEV in the “over 65 years” group could be justified by its good tolerability in older people, and especially with the lack of negative effects on cognition, even in cognitive-impaired subjects [55]; conversely, its use in younger patients generates more concern for the risk of behavioral and psychological changes.

Then, we compared the mean C(p) of the different ASMs between females and males, but in most cases, we did not find any particular difference in relation to gender. Concerning age, for VPA, this was significantly higher for women compared to men (48.9 vs. 40.2 years), with a relevant percentage of female patients in the “over 65 years” group (57%). This might reflect VPA restrictions in fertile females for its well-known teratogenic effect and potential cognitive effects in the offspring. On the contrary, women whose C(p) of LTG and OxCBZ were assayed were younger than men, reflecting the recent indications by large pregnancy outcomes international registries that these two may represent good alternatives to VPA for women at childbearing age, especially when administered at relatively low doses [56,57,58].

The above evaluations analyzed the general use of ASMs in clinical practice, without discriminating if they were utilized for psychiatric or neurological disorders. Thus, as a further investigation, we decided to compare the C(p) samples of the four ASMs that are used both in neurologic and psychiatric patients, i.e., VPA, CBZ, OxCBZ, and LTG, fairly assuming that the request form coming from the Neurology or Psychiatry Unit could distinguish between these two categories. However, we should note that samples coming from the Neurology Unit could be referring to medical conditions other than epilepsy (e.g., VPA or TPM for migraine and TPM for neuropathic pain), even if epilepsy was certainly largely represented in the vast majority of the neurological samples.

From this comparison, we found a lower mean C(p) for the ones requested by the Psychiatric Unit compared to the Neurology Unit, with the exception of VPA. These ASMs are used as mood stabilizers in BD, and the lower C(p) values could be the consequence of the frequent use of multiple medications in psychiatric disorders, which might persuade the psychiatrist to use lower dosages, and/or the difficulty adhering to the therapy for BD patients [59]. For the Child Neuropsychiatric Unit, which includes either neurological or psychiatric patients, the C(p) was more similar to the Neurology Unit, with the exception of VPA.

On the contrary, when we analyzed VPA samples, we found a slightly higher C(p) in samples coming from the Psychiatry Unit compared to those from the Neurology Unit (51 and 45 μg/mL, respectively). One possible explanation is that patients of the Neurology Unit had a mean age that was higher than the psychiatric ones, suggesting a more cautious approach in the use and titration of ASMs in the elderly.

A relevant number (1688) of patients were retested for their ASMs C(p) during this period, and we found that these values in general remained stable during a period of 1 year, and the mean C(p) was kept in the reference range, which might be in line with the good compliance towards these medications. The only exception was represented by TPM, for which the mean initial C(p) above the therapeutic range was slightly reduced thereafter. In addition, when we considered ASMs (VPA, LEV, LTG, and ESM) subjected to multiple dosing starting from an initial sub-therapeutical C(p), we observed a clear increase of C(p) values throughout the 12-month period, which is in line with the scope of their repeated assay specifically to monitor therapeutic levels for ASMs.

As a final part of our study, we searched for ASMs combination therapies, since it is known that in up to 1/3 of epileptic patients, monotherapy does not lead to seizure freedom, and a common approach in the DRE is to add on diverse ASMs [24].

About 10% of the patients of our sample were taking 2 or more ASMs, which were monitored at the same time; however, this number is probably an underestimation, since it may not include those subjects in which, even though the ASMs were administered in association, their C(p) were tested at different times.

The most common combinations were LEV/VPA, ESM/VPA, CBZ/VPA, and CBZ/LEV. These associations seem reasonable in terms of the different mechanism of actions and increased risk of side effects; however, they impose a more stringent monitoring of patients’ conditions and pharmacokinetic interactions that could alter the ASMs C(p) [60].

In conclusion, this study has shown that most of the ASM C(p) was within the reference ranges, reflecting an overall proper use of ASMs in clinical practice in the west-Tuscany health system, and especially in the Pisa University Hospital Neurology/Psychiatry/Child Neuropsychiatry Units. If, as one may expect, the therapeutic approaches used in these centers are in line with those of most Italian regions, this might indicate that these drugs are nowadays quite appropriately used alone, or in combination. VPA represented, at a first sight, an exception, since it that was found to be underdosed in several samples. However, this may reflect its current wide and versatile use in diverse neuropsychiatric disorders, either as monotherapy or in association with other drugs. Importantly, toxic levels of ASMs were very rare in our series, and when sub-therapeutical C(p) was detected, an increase of C(p) values was found in subsequent measurements, which is a demonstration of TDM utility in dosage change during the treatment.

This study has been performed in a purely “naturalistic” setting, without clinical information about the patients, in which we focused exclusively on ASMs plasma level concentrations in a centralized lab collecting samples from different clinical sources. The daily dosages of the patients were not available on the request form and thus were not used, and this is an important limitation that needs to be mentioned. We also acknowledge as another potential limitation the possible use of other pharmacotherapies in combination with ASMs that have not been monitored.

Indeed, this is a first retrospective study based on a very large number of ASMs laboratory exams, which should be followed by a prospective clinical trial trying to correlate the patient clinical response to ASMs C(p).

## 4. Materials and Methods

### 4.1. Study Sample

ASMs blood samples (C(p)) were collected and analyzed from 2018 to 2022 at the Unit of Pharmacology and Pharmacogenetics of the Hospital of University of Pisa (Azienda Ospedaliero-Universitaria Pisana, AOUP). The total number of C(p) levels was 23,946, 1036 of which came from Neurology Units, 6064 came from Psychiatry Units, 4092 came from Child Neuropsychiatry Units, and 12,753 came from other clinical units. Other units included internal and external requests. External request samples were collected from both hospitals and Territorial Health Departments in the Provinces of Pisa, Lucca, Livorno, and Massa-Carrara, Tuscany, Italy (Area nord-ovest Toscana). We included all ASMs with at least 100 C(p) samples in order to be able to provide proper statistical analyses for our data (i.e., for zonisamide, we only found 3 blood samples in the 5-year period). We did not apply any exclusion criteria for patients C(p) that were collected as a request form using open Lab Information System (OpenLis) software. Samples values not detectable were not included in the statistical analyses.

### 4.2. Analytical Determination of ASMs Plasma Concentrations

Samples were collected, following routine clinical practice, every morning from patients before taking the next dose (at steady state), and then stored in the refrigerator at 4 °C.

VPA concentrations were measured with the ARCHITECT iValproic Acid assay, a chemiluminescent microparticle immunoassay. Plasma samples (150 µL each) were combined with anti-valproic acid-coated paramagnetic microparticles and valproic acid acridium-labeled conjugate in order to create a reaction mixture. The resulting chemiluminescent reaction was measured as relative light units. The LoB and LoD of the ARCHITECT iValproic Acid assay were determined, according to the CLSI Protocol EP17-A21, with a LoB = 0.27 μg/mL and LoD = 0.51 μg/mL.

LEV concentrations were determined with HPLC (high-performance liquid chromatography) with UV detection. Plasma was separated by red blood cells by centrifugation at 400 rpm (5 min); samples were prepared according to the 24,000 Chromsystems ^®^ Kit (Chromsystems Instruments & Chemicals GmbH, Gräfelfing/Munich, Germany) procedure, and instrumental parameters were set as per the manufacturer’s instructions. Isocratic elution was performed with an injection volume of 10 µL and a flow rate of 1.5 mL/min, and the ultraviolet detection was at 210 nm. The LoQ of the Levetiracetam HPLC assay was 0.5 mg/L, with linearity up to 1000 mg/L. The analysis time was 6 min per sample.

OxCBZ, 10-OH-CBZ, CBZ-10,11-epoxide, LTG, ESM, PHT, and PB were analyzed by the 22,000 Chromsystems™ kit. After centrifugation, clear supernatants were injected into the high-performance liquid chromatography system. Isocratic elution was performed with an injection volume of 20 µL and a flow rate of 1.2 mL/min, and the ultraviolet detection was at 204 nm. The method presents a linearity for the entire therapeutical range, a limit of detection of 0.5 mg/L, an intraassay CV < 4%, an interassay CV < 6%, a recovery of 90%, and a run time (high resolution) of 20 min.

LCS was assessed by the 21,000 Chromsystems™ kit; isocratic elution was performed with an injection volume of 20 µL and a flow rate of 2.5 mL/min, and the ultraviolet detection was at 204 nm. The method presents a linearity for the entire therapeutical range, a limit of detection equal to 0.5 mg/l, an intraassay CV < 2%, an interassay CV < 2%, a recovery of 100%, and a run time (high resolution) of 15 min.

Regarding TPM analysis, the quantification was performed by a turbidimetric method developed for the Indiko™ Clinical Chemistry Analyzer (ThermoFisher™ Scientific Inc., Waltham, MA, USA), using the ARK™ Topiramate Assay (catalog #5015-0001-00; ARK diagnostic Inc., Fremont, CA, USA). This method presents a linearity up to 54.0 μg/mL, a limit of detection equal to 1.5 μg/mL, an intraassay CV < 3.4%, an interassay CV < 4.3%, and a recovery >95% at the concentration of 1.5 μg/mL.

The results were then registered in the Openlis software, a database accessed by the University Hospital of Pisa by using a card for recognition with a password in order to protect the privacy of the patients. Our laboratory activity is ISO9001-certified, and it is routinely tested with external samples for quality control.

### 4.3. Data Analysis

We reported data samples for different subgroups (years, origin, range intervals, age). C(p) across different ages was reported as mean, and Kruskal–Wallis or ANOVA tests were performed according to each ASM distribution, after performing the Kolmogorov–Smirnov normality test. If only two groups were present, we used the Mann–Whitney U test. Correlations were calculated with the Spearman r test. We used the Friedman test to analyze repeated measures of C(p)s in the same patient across a 12-month period. The level of significance was set at 0.05. The graphs; descriptive analysis; ANOVAs; and Kruskal–Wallis, Mann–Whitney, Spearman r, and Friedman tests were made with GraphPad Prism 7.05 for Windows (GraphPad Software, San Diego, CA, USA).

### 4.4. Ethical Statement

All patients gave their consent to the collection of TDM analyses and the subsequent use of their data for research. The data were anonymous when collected for the different analyses. The study was conducted in accordance with the Declaration of Helsinki, and the protocol was approved by the Ethics Committee of the Area Vasta Nord Ovest (CEAVNO) (Project identification code # CEAVNO_Scarselli_24-05-2023).

## Figures and Tables

**Figure 2 pharmaceuticals-16-00945-f002:**
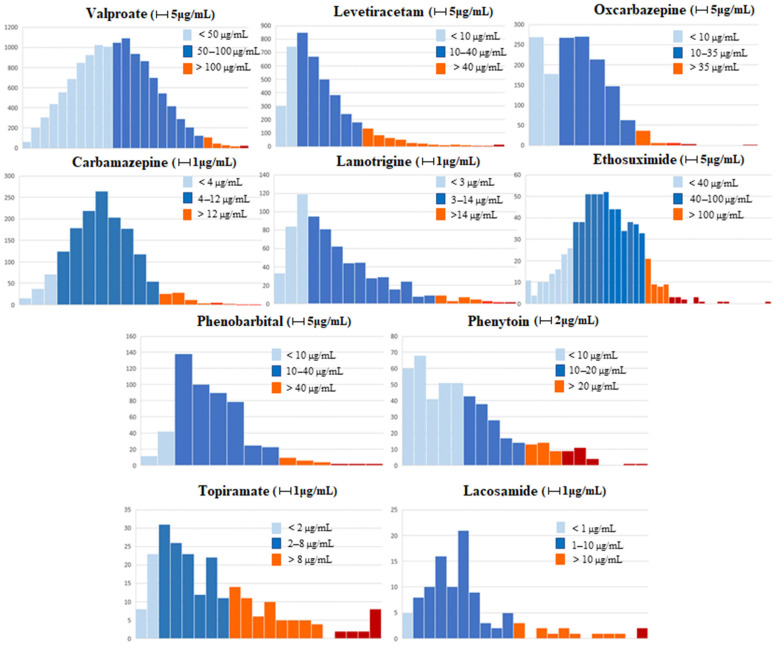
ASMs C(p) blood samples distribution in relation to their reference ranges. Light blue color Light blue color (
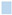
) indicates under-range dosage, blue (🟦) in range, orange (🟧) over-range, and red (🟥) toxic levels. VPA, LEV, OxCBZ, ESM, and PB intervals are 5 μg/mL; PHT intervals are 2 μg/mL; CBZ, LTG, TPM, and LCS intervals are 1 μg/mL.

**Figure 3 pharmaceuticals-16-00945-f003:**
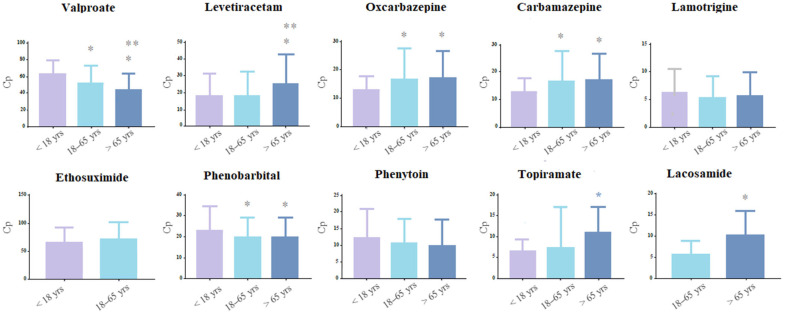
ASMs distribution across age divided in three groups: under 18 years, between 18–65 years, and over 65 years. For the statistical comparison among the three groups, the first group was used as reference. * statistically significant difference compared to the first age group. ** statistically significant difference between the third and the second age group. For VPA *p* < 0.001, F = 12.88; LEV *p* < 0.001, H = 63.73; TPM *p* < 0.001, H = 59.92; LCS *p* < 0.001, U = 139. For OxCBZ *p* = 0.004, H = 8.46; for CBZ *p* = 0.008, F = 4.88; for PB *p* = *0*.02, F = 2.29.

**Figure 4 pharmaceuticals-16-00945-f004:**
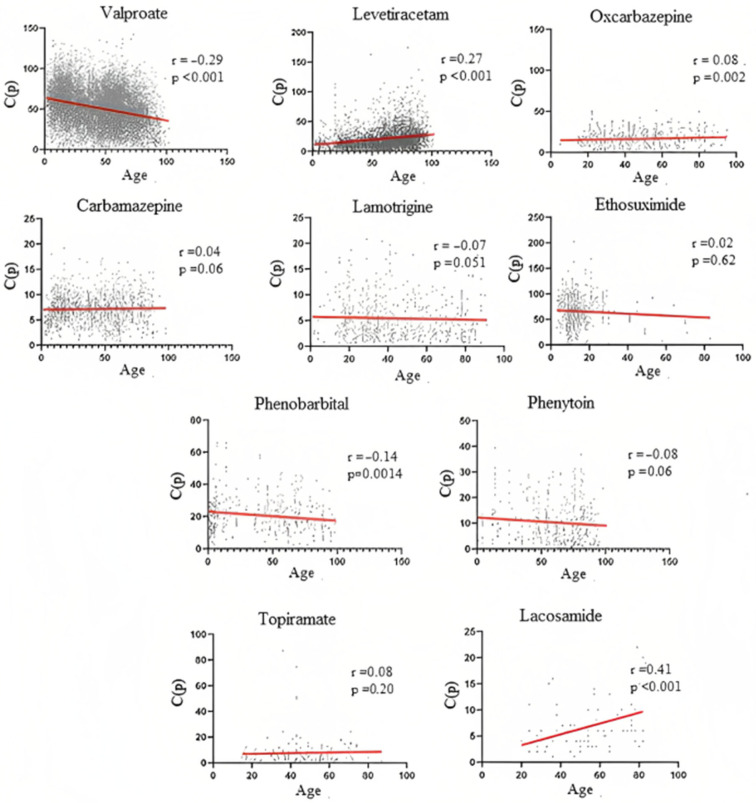
Correlation between age and C(p) distribution among the different ASMs. The number of samples refers to the ones previously indicated in Figure 1A.

**Figure 6 pharmaceuticals-16-00945-f006:**
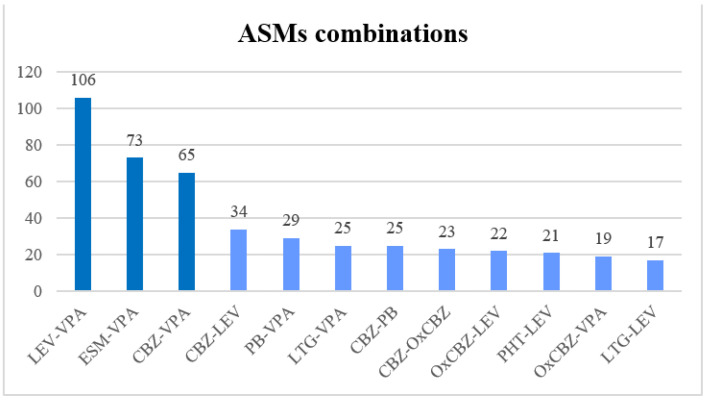
The most common combinations between two ASMs found in our samples. Abbreviations: VPA—valproate, LEV—levetiracetam, ESM—ethosuximide, CBZ—carbamazepine, PB—phenobarbital, LTG—lamotrigine, OxCBZ—Oxcarbazepine, PHT—phenytoin.

**Table 1 pharmaceuticals-16-00945-t001:** C(p) (mean) with standard deviations (SDs) of ASMs and their percentage in range, under-range, and over-range.

ASM (Reference Range)	C(p) (Mean)	In Range	Under-Range	Over-Range
Valproate (50–100 μg/mL)	51.3 ± 22.2	49.8%	48.4%	1.8%
Levetiracetam (10–40 μg/mL)	21.2 ± 16.3	65.5%	23.7%	10.8%
Oxcarbazepine (10–35 μg/mL)	16.3 ± 10.6	68.5%	27.6%	3.9%
Carbamazepine (4–12 μg/mL)	7.14 ± 2.67	76.4%	17.9%	3.7%
Lamotrigine (3–14 μg/mL)	5.4 ± 3.9	64.1%	32.8%	4.4%
Ethosuximide (40–100 μg/mL)	65.8 ± 27.6	74.3%	16.6%	9%
Phenobarbital (10–40 μg/mL)	20.5 ± 10.5	85.6%	9.5%	4.9%
Phenytoin (10–20 μg/mL)	10.3 ± 7.8	56.7%	30.2%	13.1%
Topiramate (2–8 μg/mL)	7.6 ± 9.4	55.2%	12.6%	32.2%
Lacosamide (1–10 μg/mL)	6.6 ± 4.4	85.1%	0%	13.9%

**Table 2 pharmaceuticals-16-00945-t002:** Mean age and ASMs C(p) with SD in females and males, while the number of patients for each group is indicated in parentheses. For age comparison between females and males, * indicates a *p* < 0.001 for VPA, LEV, OxCBZ, and PB; *p* = 0.008 for LTG; *p* = 0.02 for LCS, whereas for C(p) differences in gender, § represents *p* < 0.001 for LEV, *p* = 0.004 for OxCBZ, and *p* = 0.01 for CBZ.

ASM (Reference Range)	Female Mean Age	Male Mean Age	Females Mean C(p)	Males Mean C(p)
Valproate (50–100 μg/mL)	48.9 ± 23.6 (2236)	40.2 ± 22.5 * (2714)	51.7 ± 22.6	51.05 ± 21.9
Levetiracetam (10–40 μg/mL)	62.2 ± 22.1 (946)	60.1 ± 20.7 * (880)	22.9 ± 18	19.5 ± 14.2 §
Oxcarbazepine (10–35 μg/mL)	44.6 ± 19.5 (190)	49.3 ± 20.1 * (134)	15.6 ± 10.7	17.2 ± 10.3 §
Carbamazepine (4–12 μg/mL)	40.5 ± 24.2 (357)	38.9 ± 24 (362)	6.97 ± 2.6	7.3 ± 2.8 §
Lamotrigine (3–14 μg/mL)	39.6 ± 19.7 (211)	48.2 ± 22.9 * (99)	5.4 ± 3.5	5.5 ± 4.7
Ethosuximide (40–100 μg/mL)	14.4 ± 9.8 (109)	13.9 ± 9.7 (93)	67.4 ± 28.7	64 ± 26.2
Phenobarbital (10–40 μg/mL)	36.7 ± 32.2 (108)	46.4 ± 27.2 * (119)	19.8 ± 9.4	21 ± 11.1
Phenytoin (10–20 μg/mL)	61 ± 25.5 (89)	59.2 ± 22.9 (105)	9.9 ± 7.9	10.5 ± 7.8
Topiramate (2–8 μg/mL)	47.8 ± 17.5 (60)	44.8 ± 14.4 (52)	7.9 ± 9.8	7.2 ± 8.9
Lacosamide (1–10 μg/mL)	56.9 ± 18.1 (37)	49 ± 16.8 * (33)	7.2 ± 5.6	6.1 ± 2.9

**Table 3 pharmaceuticals-16-00945-t003:** C(p) (mean) with SD across different clinical units: Neurological, Psychiatric, and Child Neuropsychiatry Units. The number of samples is indicated in parentheses. For the statistical comparison between the three groups, the first group was used as reference. * indicates for VPA *p* < 0.001, F = 18.84; for CBZ *p* < 0.001, F = 1.28; for OxCBZ *p* < 0.001, H = 7.81; and for LTG *p* = 0.01, H = 0.036. ** indicates statistically significant differences between the second and the third groups (VPA: *p* < 0.001, CBZ: *p* = 0.02).

ASM (Reference Range)	Psychiatry Unit Mean C(p)	Neurology Unit Mean C(p)	Child Neuropsy Unit Mean C(p)
Valproate (50–100 μg/mL)	51 ± 21.2 (5191)	45 ± 22.4 * (240)	60.9 ± 20.6 *, ** (2627)
Oxcarbazepine (10–35 μg/mL)	13.7 ± 8.1 (320)	21.3 ± 13 * (27)	18 ± 11.6 * (123)
Carbamazepine (4–12 μg/mL)	6.6 ± 2.6 (182)	9.1 ± 3.8 * (30)	7.1 ± 2.5 ** (474)
Lamotrigine (3–14 μg/mL)	2.5 ± 1.7 (12)	4.7 ± 4.8 (26)	5.3 ± 3.5 * (59)

## Data Availability

Data are contained within the article.

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
