# Peer review of "A 5-Year Study of Antiseizure Medications (ASMs) Monitoring in Patients with Neuropsychiatric Disorders in an Italian Clinical Center"

_pharmaceuticals, 2023, doi:10.3390/ph16070945_

Round 1

Reviewer 1 Report

This manuscript describes a large study of therapeutic drug monitoring for seizure medications. The paper concludes that in general, levels of seizure medications are within normal range. While the results are somewhat interesting, they overall do not add much to the existing understanding of this topic. Further, there are some significant methodological issues with this study. Specific comments are below:

1. Recommend using current terminology (ASM instead of AED)

2. Overall, the paper is extremely confusing - starting with the abstract and introduction in terms of exactly what diagnosi(e)s the patients studied have. In general the sample seems extremely heterogenous in regards to diagnosis but there is no confirmation of this. There is implication at times that all of the patients have epilepsy, but then at other times the broad term of "neuropsychiatric disorders" is used and comparison is made between neurology units and psychiatric units. This is a major weakness of this paper. Dosages of ASMs used to treat psychiatric conditions are different than they would be to treat epilepsy. Further, ASMs often have other indications (in particular migraine and sometimes neuropathic pain). There needs to be some mention of diagnosis of these patients. The conclusions as a whole strictly relate to epilepsy but it is not clear if all of the patients studied have an epilepsy diagnosis. 

3. Methods section should be moved to after introduction and before results. Further, mention of whether or not these samples were appropriately drawn ie trough levels is not made. The assumption is not, but this is a critical point made in the introduction regarding the current data available and thus some mention should be made of this.

4. Please include what reference ranges were used as "normal range" for drug levels

5. There are a few mentions which are assumed to be describing the definition of drug refractory epilepsy; however, the statements presented do not appropriately define DRE.. There is one sentence in the introduction and one in the discussion section which need to be corrected.

6. Was dose of ASM collected as part of this analysis? Was this controlled for in the analysis? If not, this needs to be mentioned specifically as a limitation

There were some minor issues with English language (use of commas in particular), but overall minor.

Author Response

Referee #1

This manuscript describes a large study of therapeutic drug monitoring for seizure medications. The paper concludes that in general, levels of seizure medications are within normal range. While the results are somewhat interesting, they overall do not add much to the existing understanding of this topic. Further, there are some significant methodological issues with this study. Specific comments are below:

Reply:

We thank the reviewer for his/her comment finding the results interesting and we tried to improve the manuscript following carefully all the reviewers’ comments.

  1. Recommend using current terminology (ASM instead of AED).

Reply:

Following reviewer’s advice, we replaced AEDs with antiseizure medications (ASMs) throughout the text.

2a) Overall, the paper is extremely confusing - starting with the abstract and introduction in terms of exactly what diagnosi(e)s the patients studied have. In general the sample seems extremely heterogenous in regards to diagnosis but there is no confirmation of this. There is implication at times that all of the patients have epilepsy, but then at other times the broad term of "neuropsychiatric disorders" is used and comparison is made between neurology units and psychiatric units. This is a major weakness of this paper.

Reply:

We agree with the reviewer that in several parts it was not clear to which medical condition we were referring to, if epilepsy or other neuropsychiatric disorders. Accordingly, we made several changes regarding this point (highlighted) throughout the text. In relation to the patient’s diagnosis, we did not have clinical information about the patients but in the TDM request form it was indicated the name of the Unit (e.g. Neurology or Psychiatry). However, now we have underlined even more than before that this was a limitation of this study (as suggested by the reviewer later). We also would like to say that according to the recent consensus guidelines for TDM in neuropsychopharmacology (Hiemke C. et al 2018), the same reference ranges for ASMs plasma concentrations (TDM) in epilepsy and bipolar disorders have been adopted.

We show here some examples of changes have been made:

Abstract:

- “Despite receiving appropriate antiseizure medications (ASMs), a relevant percentage of neuropsychiatric patients do not benefit from this approach and one reason is subtherapeutic..”

- “Based on these premises, in the present real-world study, we analysed the C(p) of the most used ASMs in diverse medical conditions.....We did not have any clinical information about the patients, so the sample was largely heterogeneous…. This population was largely heterogeneous and our database did not contain clinical information about the patients.”

- “Then, comparing ASMs used in neurologic and psychiatric disorders, based on the request form,”

Introduction:

- “Based on these premises, in this study, we analyzed the C(p) of the most used ASMs in diverse medical conditions during the years 2018– 2022 at the University Hospital of Pisa, including about 24,000 samples. We did not have clinical information about the patients, so the sample was largely heterogeneous…..Finally, we tried to compare the C(p) used in neurologic and psychiatric patients, based on the request form coming from the Neurology or Psychiatry Unit that allowed us to make this comparison.”

Discussion:

- “In our study, we analyzed the results of plasmatic levels of the ten most used ASMs in diverse neuropsychiatric conditions”

- “The above evaluations analyzed the general use of ASMs in clinical practice without discriminating if they were utilized for psychiatric or neurological disorders. Thus, as a further investigation, we decided to compare the C(p) samples of the four ASMs that are used both in neurologic and psychiatric patients, i.e., VPA, CBZ, OxCBZ, and LTG, fairly assuming that the request form coming from the Neurology or Psychiatry Unit could distinguish between these two categories. However, we should remind that samples coming from the Neurology Unit could be referring to medical conditions other than Epilepsy (e.g. VPA or TPM for migraine and TPM for neuropathic pain), even if Epilepsy was certainly largely represented in the vast majority of the neurological samples.”

2b) Dosages of ASMs used to treat psychiatric conditions are different than they would be to treat epilepsy. Further, ASMs often have other indications (in particular migraine and sometimes neuropathic pain). There needs to be some mention of diagnosis of these patients. The conclusions as a whole strictly relate to epilepsy but it is not clear if all of the patients studied have an epilepsy diagnosis. 

Reply:

- As mentioned by the reviewer, ASMs can be used at different dosage according to the medical condition (we added this in the text), even if studies considering different ASMs dosages in psychiatric disorders or neurological disorders other than epilepsy are heterogeneous and have proposed a wide dose-range (Smits et al., 2017; Unholzer & Haen, 2015; Hiemke et al., 2018). However, as already said, according to the recent consensus guidelines for TDM in neuropsychopharmacology (Hiemke C. et al 2018), the same reference ranges for ASMs plasma concentrations (TDM) in epilepsy and bipolar disorders have been adopted. In addition, data on TDM for ASMs in migraine (mostly VPA and TPM) and neuropathic pain (mostly gabapentin that we did not find in our analysis) are still missing (Parikh & Silberstein, 2019), so this limits their plasma monitoring in clinical practice, particularly, in the Italian Ministry of Health system. In relation to the patient’s diagnosis, we did not have clinical information about patients but in the TDM request form it was indicated the name of the Unit that allowed us to distinguish at least between neurological and psychiatric patients. However, we underlined that this was an important limitation that was mentioned at the end of the discussion (as suggested by the reviewer later). For the possibility of patient’s samples coming from the Neurology Unit and referring to medical conditions other than Epilepsy: we also added this point. Accordingly, we made several changes regarding this point (highlighted) throughout the text and we show here some examples:

Abstract:

- “This population was largely heterogeneous and our database did not contain clinical information about the patients”.

Introduction:

- “ASMs are generally used in all the different forms of epilepsy, but some of them are also frequently used for other disorders, primarily in bipolar disorders and secondary in migraine and neuropathic pain [8–11]. Dosages of ASMs can be different according to the medical condition (Hiemke et al., 2018; Freitag et al., 2002; Smits et al., 2017). Importantly, the reference ranges of ASMs C(p) were determined in epileptic patients, whereas TDM studies for the other medical conditions, such as bipolar disorders, are less numerous (Smits et al., 2017; Talaei et al., 2022; Unholzer & Haen, 2015). However, according to the recent consensus guidelines for TDM in neuropsychopharmacology, the same reference ranges of ASMs for epilepsy and bipolar disorder have been adopted (Hiemke et al., 2018). Studies on TDM for ASMs used in migraine or neuropathic pain are still missing, limiting their use in clinical practice, so we can assume that in our sample it was less represented (Freitag et al., 2002; Gill et al., 2011).”

Discussion:

- “In addition, VPA can be also used in migraine, however data of TDM for this medical condition are still missing, so we can assume that in our sample it was less represented (Linde et al., 2013).”

- “The above evaluations analyzed the general use of ASMs in clinical practice without discriminating if they were utilized for psychiatric or neurological disorders. Thus, as a further investigation, we decided to compare the C(p) samples of the four ASMs that are used both in neurologic and psychiatric patients, i.e., VPA, CBZ, OxCBZ, and LTG, fairly assuming that the request form coming from the Neurology or Psychiatry Unit could distinguish between these two categories. However, we should remind that samples coming from the Neurology Unit could be referring to medical conditions other than Epilepsy (e.g. VPA or TPM for migraine and TPM for neuropathic pain), even if Epilepsy was certainly largely represented in the vast majority of the neurological samples.”

- “This study has been performed in a purely “naturalistic” setting without clinical information about the patients, in which we focused exclusively on ASMs plasma level concentrations in a centralized Lab collecting samples from different clinical sources. The daily dosages of the patients were not available in the request form, and thus were not used, and this is a relevant limitation of this study. We also acknowledge as another potential limitation the possible use of other pharmacotherapies in combination with ASMs that have not been monitored.”

  1. Methods section should be moved to after introduction and before results. Further, mention of whether or not these samples were appropriately drawn ie trough levels is not made. The assumption is not, but this is a critical point made in the introduction regarding the current data available and thus some mention should be made of this.

Reply:

The journal “Pharmaceuticals” generally places the Methods section at the end. However, if the Editor and the reviewer agree on change this, we can proceed with this different approach.

To clarify the part in relation to the samples and their levels, we propose as follow:

Materials and Methods:

- “ASMs blood samples (C(p)) were collected and analysed from 2018 to 2022 at the Unit of Pharmacology and Pharmacogenetics of the Hospital of University of Pisa (Azienda Ospedaliero-Universitaria Pisana, AOUP). The total number of C(p) levels was 23946, 1036 of which came from Neurology Units, 6064 came from Psychiatry Units, 4092 came from Child Neuropsychiatry Units, and 12753 came from other clinical Units. Other units included internal and external requests. External request samples were collected from both Hospitals and Territorial Health Departments in the Provinces of Pisa, Lucca, Livorno, and Massa-Carrara, Tuscany, Italy (Area nord-ovest Toscana). We included all ASMs with at least 100 C(p) samples, in order to be able to provide proper statistical analyses to our data (i.e., for zonisamide we only found 3 blood samples in the 5-year period). We did not apply any exclusion criteria for patients C(p) which were collected as request form using open Lab Information System (OpenLis) software. Samples values not detectable were not included in the statistical analyses.”

  1. Please include what reference ranges were used as "normal range" for drug levels

Reply:

- The reference ranges were indicated in all the tables according to the recent consensus guidelines for TDM in neuropsychopharmacology (Hiemke C. et al 2018). To make them more visible, we put them in bold character.

  1. There are a few mentions which are assumed to be describing the definition of drug refractory epilepsy; however, the statements presented do not appropriately define DRE. There is one sentence in the introduction and one in the discussion section which need to be corrected.

Reply:

As suggested by the reviewer, we corrected this information.

Introduction:

“Indeed, polytherapy, i.e., the use of two or even three ASMs in the same individual, is not rarely used in the clinical practice for patients with drug-refractory epilepsy (DRE), which is a form of epilepsy with uncontrolled seizures that persists after the treatment with at least two appropriately chosen and administered ASMs [13,19] (Kwan et al., 2010; Kramer 2022).”

Discussion:

“As a final part of our study, we searched for ASMs combination therapies, since it is known that in up to 1/3 of epileptic patients monotherapy does not lead to seizure-freedom, and a common approach in the DRE is to add-on diverse ASMs (24).”

  1. Was dose of ASM collected as part of this analysis? Was this controlled for in the analysis? If not, this needs to be mentioned specifically as a limitation.

Reply:

As we already said, the dose of the ASMs was not available on our data base, so we mentioned at the end as an important limitation, as suggested by the reviewer.

Discussion:

- “This study has been performed in a purely “naturalistic” setting without clinical information about the patients, in which we focused exclusively on ASMs plasma level concentrations in a centralized Lab collecting samples from different clinical sources. The daily dosages of the patients were not available in the request form, and thus were not used, and this is an important limitation that needs to be mentioned“.

Reviewer 2 Report

I find the questions addressed in the article to be of potential interest, however there are some corrections that need to be made. 

The abbreviation “Cp” is associated with ceruloplasmin, in my opinion.

There is a repeat of the description on pages 116-119, which is shown., in the methods, it should be removed.

 In  Fig 1 the label for the y-axis is not readable, it should be moved.

In Fig 2  the interval value should be  indicated under each histogram.

Fig 3 duplicates the table, which is betteг taken. Thus, I guess, the picture should be removed.

The volume of VPA and LEV groups is very large, so, it is better to use uniformity criteria

for the comparison. Also it is required to specify the F-value for each comparison.

Fig 4 – Quality of the picture is low.

Tab 4 – It is  necessary to point  F - value for each comparison.

Author Response

Referee #2

I find the questions addressed in the article to be of potential interest, however there are some corrections that need to be made. 

Reply:

We thank the reviewer for his/her comment on the article and for his/her fruitful comments, and we tried to improve the manuscript following carefully all the reviewers’ comments.

- The abbreviation “Cp” is associated with ceruloplasmin, in my opinion.

Reply:

Following reviewer’s advice, we changed the abbreviation and now we use C(p).       .

- There is a repeat of the description on pages 116-119, which is shown., in the methods, it should be removed.

Reply:

We removed this part.

- In  Fig 1 the label for the y-axis is not readable, it should be moved.

Reply:

We put the label for y-axis in vertical, so now it is better visualized.

- In Fig 2  the interval value should be indicated under each histogram.

Reply:

We indicated the value in the figure.

- Fig 3 duplicates the table, which is betteг taken. Thus, I guess, the picture should be removed.

The volume of VPA and LEV groups is very large, so, it is better to use uniformity criteria

for the comparison. Also it is required to specify the F-value for each comparison.

Reply:

We agree with the reviewer that Table 2 and Figure 3 are redundant showing the same data. For this reason, we decided to eliminate the table and keep the figure, which visualizes better the differences, as our personal opinion. We added the F-value or H-value for each comparison when needed. For the uniformity criteria, considering we analyzed all the ASMs separately comparing them with other variables, we prefer for the moment not to go into a more detailed statistical analysis for VPA and LEV hoping that the reviewer will find these statistical changes we made enough for the purpose of the article.

Fig 4 – Quality of the picture is low.

Reply:

We improved the quality of this figure.

Tab 4 – It is  necessary to point  F - value for each comparison.

Reply:

We added the F and H values.

Reviewer 3 Report

The authors report data about plasma concentration of the main antiseizure medications in a large sample; this analysis is really well done and interesting for the clinicians. Overall the manuscript is well organized. Therefore, I have few comments for the authors:

- The term antiepileptic drugs AEDs is no longer in use and must be replaced with antiseizure medications (ASMs) throughout the text.

- The authors should add more details about how they recruited and selected their patients (e.g. inclusion and exclusion criteria).

- Discussion: the authors sould underline that, sometime, some types of epilepsy can get complicated by neurophycological problems: the authors must read and cite the paper by Verrotti A et al. Journal of the Neurological Sciences, 2015; 359(1-2), 59–6.

Author Response

Referee #3

The authors report data about plasma concentration of the main antiseizure medications in a large sample; this analysis is really well done and interesting for the clinicians. Overall the manuscript is well organized. Therefore, I have few comments for the authors:

Reply:

We thank the reviewer for his/her positive comment on the article underling that the paper is well done, interesting and well organized.

- The term antiepileptic drugs AEDs is no longer in use and must be replaced with antiseizure medications (ASMs) throughout the text.

Reply:

Following reviewer’s advice, we replaced AEDs with antiseizure medications (ASMs) throughout the text.

- The authors should add more details about how they recruited and selected their patients (e.g. inclusion and exclusion criteria).

Reply:

We added more information in relation to inclusion criteria in the Materials and Methods section, as follow:

Materials and Methods:

- “ASMs blood samples (C(p)) were collected and analysed from 2018 to 2022 at the Unit of Pharmacology and Pharmacogenetics of the Hospital of University of Pisa (Azienda Ospedaliero-Universitaria Pisana, AOUP). The total number of C(p) levels was 23946, 1036 of which came from Neurology Units, 6064 came from Psychiatry Units, 4092 came from Child Neuropsychiatry Units, and 12753 came from other clinical Units. Other units included internal and external requests. External request samples were collected from both Hospitals and Territorial Health Departments in the Provinces of Pisa, Lucca, Livorno, and Massa-Carrara, Tuscany, Italy (Area nord-ovest Toscana). We included all ASMs with at least 100 C(p) samples, in order to be able to provide proper statistical analyses to our data (i.e., for zonisamide we only found 3 blood samples in the 5-year period). We did not apply any exclusion criteria for patients C(p) which were collected as request form using open Lab Information System (OpenLis) software. Samples values not detectable were not included in the statistical analyses.”

- Discussion: the authors sould underline that, sometime, some types of epilepsy can get complicated by neurophycological problems: the authors must read and cite the paper by Verrotti A et al. Journal of the Neurological Sciences, 2015; 359(1-2), 59–6.

Reply:

We added this part and the citation in the Discussion.

Round 2

Reviewer 1 Report

All comments addressed.